# It Takes Two to Tango: Endothelial TGFβ/BMP Signaling Crosstalk with Mechanobiology

**DOI:** 10.3390/cells9091965

**Published:** 2020-08-26

**Authors:** Christian Hiepen, Paul-Lennard Mendez, Petra Knaus

**Affiliations:** Knaus-Lab/Signal Transduction, Institute for Chemistry and Biochemistry, Freie Universitaet Berlin, 14195 Berlin, Germany; christian.hiepen@fu-berlin.de (C.H.); pmendez@zedat.fu-berlin.de (P.-L.M.)

**Keywords:** BMP, TGFβ, mechanobiology, endothelial-cell, shear stress

## Abstract

Bone morphogenetic proteins (BMPs) are members of the transforming growth factor-beta (TGFβ) superfamily of cytokines. While some ligand members are potent inducers of angiogenesis, others promote vascular homeostasis. However, the precise understanding of the molecular mechanisms underlying these functions is still a growing research field. In bone, the tissue in which BMPs were first discovered, crosstalk of TGFβ/BMP signaling with mechanobiology is well understood. Likewise, the endothelium represents a tissue that is constantly exposed to multiple mechanical triggers, such as wall shear stress, elicited by blood flow or strain, and tension from the surrounding cells and to the extracellular matrix. To integrate mechanical stimuli, the cytoskeleton plays a pivotal role in the transduction of these forces in endothelial cells. Importantly, mechanical forces integrate on several levels of the TGFβ/BMP pathway, such as receptors and SMADs, but also global cell-architecture and nuclear chromatin re-organization. Here, we summarize the current literature on crosstalk mechanisms between biochemical cues elicited by TGFβ/BMP growth factors and mechanical cues, as shear stress or matrix stiffness that collectively orchestrate endothelial function. We focus on the different subcellular compartments in which the forces are sensed and integrated into the TGFβ/BMP growth factor signaling.

## 1. Introduction

Since their discovery as cytokines extractable from bone matrix, data on pleiotropic activities by bone morphogenetic proteins (BMPs) belonging to the ligand superfamily of transforming growth factors beta (TGFβs) are vastly expanding. There are more than 30 known TGFβ/BMP ligands, which bind to the heteromeric receptor complexes, comprising two type I (R1) and two type II (R2) serine/threonine kinase receptors. Numerous co-receptors fine-tune signaling of these receptors [1,2,3,4]. Activated TGFβ/BMP receptor complexes signal via mothers against decapentaplegic homologs (SMADs) transcription factors or induce a number of non-canonical responses, including activation of mitogen-activated protein kinases (MAPKs), phosphoinositide-3-kinase (PI3K), as well as Rho homologous (Rho) GTPase signaling amongst others [5].

Both TGFβs and BMPs regulate important functions of the vasculature. This is underlined by several vascular phenotypes of the TGFβ/BMP-related knock-outs, which is subject to excellent reviews [6,7,8,9]. The pivotal role of these proteins for endothelial cells (ECs) forming the most inner layer of blood vessels, is further highlighted by a number of human diseases, where perturbed TGFβ/BMP signaling impedes blood vessel formation or maintenance of vascular integrity. Of note here are pulmonary arterial hypertension (PAH), Osler-Weber Rendu syndrome/hereditary hemorrhagic telangiectasia (HHT), and Loeys-Dietz-syndrome (reviewed in [9,10,11,12]).

In addition to biochemical signals, mechanical signals are equally required to orchestrate blood vessel formation, patterning, branching, pruning, and to maintain their integrity. Crosstalk of TGFβ/BMP signaling with cellular mechanobiology is a growing research field and contribution thereof to the above-mentioned vascular pathologies is little understood. In this review, we focus on the current efforts to understand how biochemical TGFβ/BMP signals in conjunction with mechanical signals, are received and integrated by the endothelium. A special focus is given to the different subcellular compartments in which we and others propose crosstalk to occur.

## 2. Activating Versus Homeostatic TGFβ/BMP Signaling in Endothelial Cells

To put endothelial mechanobiology into context, it helps to briefly introduce the different known actions of TGFβ/BMP signaling in the endothelium. ECs can transit between an activated and homeostasis state, regulated by an intricate balance between the activating and homeostatic TGFβ/BMP ligands, in addition to other potent extracellular factors [13,14,15]; Figure 1a. BMP 2, 6, and 7, all referred to as EC activating BMPs’, induce EC migration and angiogenesis [16,17,18,19,20]; Figure 1a, top. They signal mainly via R1s ALK2, ALK3, and ALK6, in conjunction with either R2s BMPR2 or ACVRIIs (Figure 1b). Sprouting angiogenesis is the formation of new blood vessels out of pre-existing ones [21]. During sprouting angiogenesis, activated tip-cells adopt a fibroblast-like front-to-rear end polarity over the course of migration. The single migratory tip-cell at sprout’s distal end is followed by multiple adjacent stalk-cells that proliferate (Figure 1a, upper) and dynamically compete with the tip-cell for its position [22]. A hallmark of tip-cell phenotype are long actin-driven filopodia at the leading edge. These protrusions are rich in integrins and form focal adhesions (FAs), connecting the extracellular matrix (ECM) with the cellular cytoskeleton, enabling the cell to sense ECM rigidity [23,24] and its nanotopography [25]. Tip-cell filopodia promote sprouting [26,27] by projecting cytokine-gradient sensing receptors to their most distal ends. Several groups have shown in vitro that different ECs, including ECs of arterial and venous origin, as well as microvascular and macrovascular ECs but also EC progenitors, respond to BMP2, 6, and 7 amongst others, by migration, sprouting, or tube-formation [17,28,29,30,31,32,33,34,35,36,37,38,39,40,41,42,43,44]. We could recently show that BMP6 promotes the expression of stalk-cell-associated genes, while BMP2 induced delta-like ligand 4 (DLL4) [17], a known tip-cell marker [45]. Application of BMP6 in a gradient-like fashion is sufficient to induce EC filopodia formation, alignment along, and chemotaxis within this gradient [17,30]. Whether EC activating BMP2, 6, 7 gradients exist in-vivo is still debatable. While BMP gradients are well-described in early developmental tissue patterning of invertebrates and vertebrates [46], their existence and contribution for sprouting angiogenesis in vivo is still not clearly shown. Mouse data on BMP-induced tumor vascularization, however suggest, that BMPs induce tumor angiogenesis, similar to vascular endothelial growth factor (VEGF)-like gradients [39,40]. Interestingly, interfering with BMP signaling in zebrafish caudal vein plexus reduces the number of tip-cells and their filopodia, by a process requiring CDC42 Rho GTPase activity [47]. Interfering with endothelial SMAD1/5 signaling in mice results in less functional tip-cells during retinal angiogenesis [48]. Taken together, EC activating BMPs 2, 6, and 7 facilitate in-vitro angiogenic sprouting through regulation of tip- and stalk-cell identities in a gradient-like fashion, and it remains to be proven if this phenomenon also occurs in vivo (Figure 1a).

TGFβ adopts a bipartite role for EC activation vs. homeostasis, dependent on its concentration and engagement of different receptor complexes. TGFβ signals via R1s ALK4 and ALK5. At early stages in the blood vessel development preceding angiogenesis, TGFβ1 mediates vasculogenesis via ALK5 (Figure 1c). Later, sprouting angiogenesis is inhibited by TGFβ1/3-ALK1/5 signaling [49,50]. Here, TGFβ signals in a so-called ‘lateral’ fashion, to activate SMAD 1/5/9 engaging ALK1 (Figure 1d). While treatment with low levels of TGFβ3 was found to inhibit proliferation and migration in mouse embryonic ECs, the opposite effect was apparent at higher concentrations [51]. This could be explained by a lateral signaling switch (Figure 1c). At higher TGFβ concentrations, ALK1-TGFβR2 complexes are activated, which transduce signals via SMAD 1/5/9, while at low levels of TGFβ, binding to the high affinity receptor complex ALK5-TβR2 is limited, which signals via SMAD 2/3 (Figure 1d). This switch in receptor recognition is reminiscent to the concentration-dependent activities of TGFβ in cancer [52]. Moreover, the EC origin/vascular bed [53] and their maturation state [49] are decisive for differential R1 expression, which might explain the bipartite pro- or anti-angiogenic activities reported for some TGFβ/BMP ligands with receptor promiscuity. Interestingly, TGFβ is stored within the extracellular matrix (ECM) (Figure 1a, middle) in a latent form, requiring integrin-dependent mechanical forces to act on its pro-domain, to be released and to activate signaling (see Section 3.1).

In sharp contrast, BMP9 and BMP10, also synthesized as large pro-domain associated precursors, freely circulate in the blood stream [54,55], while they are still associated with their pro-domains [56]. This association does not influence receptor binding [57,58,59,60]. BMP9/10 signaling provides the endothelium systemically with homeostasis/quiescent signals (reviewed in [9,19]), when angiogenic vessels become transfused with blood, e.g., after successful anastomosis [61,62]. BMP9/10 inhibit sprouting [54], promote maturation, and preserve the quiescence of ECs. In the adult lumen, the average EC divides approximately only twice in a lifetime [63]. BMP9/10 induces signaling via ALK1 (Figure 1e), the most abundant R1 expressed in ECs [59,64]. In zebrafish, it was shown that BMP9/10-Alk1 signaling limits the EC numbers and, thereby, stabilizes the caliber of nascent arteries [65]. Additionally, Alk1 expression depends on fluid shear stress (FSS) exerted by blood flow in the zebrafish [66] and some flow-responsive genes are dysregulated in Alk1 mutant arterial ECs, suggesting Alk1 to be the main BMP type I receptor integrating endothelial FSS into biochemical signaling responses [66]. Furthermore, deletion of ALK1 in mice leads to exuberated sprouting in the mouse retina [16], and addition of BMP9 normalized aberrant tumor vasculature, by decreasing permeability in Lewis lung carcinoma (mice) [67]. Studies using human cells revealed that BMP9 induces expression and secretion of stromal cell-derived factor 1 (SDF1/CXCL12), which promotes vessel maturation by regulating mural cell coverage [68], and counteracts VEGF-induced angiogenesis [59]. However, comparison of different model systems for Bmp10-Alk1 signaling should be done with care, due to the very different nature of vascular beds, flow regimes, and paralog expression [69].

While several studies report on the anti-angiogenic properties of BMP9, recent studies in human-induced pluripotent stem-cell derived ECs suggest that BMP9 also induces sprouting angiogenesis [70], which could recapitulate the above-mentioned bipartite role of TGFβ. Dependent on BMP9 concentration, receptor expression levels and the respective SMAD branches are activated.

Receptor regulated (R) SMADs consist of two domains—the MAD homology (MH) 1 domain at the amino-terminus of SMADs is important for their nuclear import and DNA binding. The C-terminal MH2 domain defines R1 binding, SMAD oligomerization, and interaction with cytosolic adaptors and transcriptional co-factors (Figure 1f, lower; exemplary canonical signaling scheme). Since R-SMADs have a low DNA binding affinity, they require the common mediator SMAD4 (co-SMAD4) and other co-factors. These co-factors integrate into SMAD complexes to regulate shuttling dynamics [71], and to make SMAD complexes competent in DNA binding and specify target gene recognition [72]. Phosphorylation of the SMAD-linker was previously shown to induce a crosstalk with the mechanical co-activator, yes-associated protein 1 (YAP1)/transcriptional co-activator with PDZ-binding motif (TAZ) signaling (see Chapter 6.3), which fine-tunes SMAD stability [73]. Additionally, SMADs compete with inhibitory SMADs (I-SMADs) for receptor and co-SMAD4 binding.

EC BMP9-SMAD 1/5 signaling upregulates about 100 genes and downregulates around 40 genes more than 2-fold [74], collectively giving rise to inhibition of ECs mitogenic response [75], while maintaining the vascular phenotype. This includes JAG1 activation and crosstalk to the NOTCH pathway by regulation of *Jagged1*, *Dll4*, Hairy/Enhancer of Split related with the YRPW motif (*HEY)* and Hairy/Enhancer of Split (*HES)* genes [76]. Additionally, BMP9 mediates EC quiescence by regulating cell-cycle-related proteins, such as c-myc and p21/waf1, and downregulates the FA complex-related genes, such as beta-actin, paxillin, and zyxin [77], suggesting BMP9 to balance EC–ECM adhesion.

Although the co-receptor Endoglin is not required for BMP9/10 activation of ALK1, it enhances its signaling output [78,79,80,81]. Endoglin binds BMP9 and potentiates BMP9-ALK1 signaling, however, it interferes with TGFβ-ALK5 signaling [82], (Figure 1c–e), thereby pushing the TGFβ/BMP balance towards BMP9 and SMAD1/5. Endoglin binds to TGFβ1,3 but not TGFβ2 [78], as well as the receptor complexes mediating signaling of Activins, BMP2, and BMP7 [79]. It also interacts with VEGF receptor 2 (VEGFR2) [83], zyxin [2], and integrins. Endoglin cooperates with FSS to potentiate BMP-induced Alk1 signaling [84], which is later explained in details. Moreover, Endoglin loss-of-function mutations mediate arterio-venous malformations found in HHT or in lung pulmonary vasculature in PAH [85]. Taken together, TGFβ/BMPs induce different functions in the activated and quiescent endothelium, which sets the base to better understand the crosstalk to mechanical cues that is introduced in the next section.

### 2.1. Mechanical Regulation of Endocytosis Suggests Modulation of TGFβ/BMP-SMAD Signaling

BMP-SMAD1/5/9 signaling, in particular its turn-over rate, is regulated by endocytosis (Figure 1f), which in turn depends on the physical properties of the plasma membrane [86,87,88]. Two major endocytic routes were described for membrane-bound BMP-receptor complexes (1) clathrin-mediated endocytosis (CME) involving clathrin-coated pits (CCPs), and (2) caveolin-dependent endocytosis [89,90]. CME facilitates SMAD signaling, as the SMAD anchor protein SARA recruits SMADs to the receptors in endosomes derived from CCPs [91,92]. Caveolin-dependent endocytosis originating from lipid rafts is involved in the regulation of non-canonical responses, including activation of MAPK [93]. The caveolae-based internalization pathway contains ubiquitin ligase Smurf2 bound receptors and was proposed to shut down the BMP/TGFβ-SMAD signaling [91], while others showed that BMP-SMAD signaling could be initiated from receptors residing within caveolae [94,95]. Importantly, physical stimuli, such as substrate stiffness sensed by ECs, were sufficient to induce caveolin-dependent endocytosis of membrane receptors [96,97,98]. Mechanistically, ALK1 and BMPR2 localize within caveolae, while Alk1 physically interacts with caveolin-1 (cav-1) through cav-1′s scaffolding domain, suggesting a joint contribution of ALK1, BMPR2, and cav-1, as the key mediators of BMP9/10 signaling [99,100,101] (Figure 1f). This is further highlighted by cav-1 mutations leading to PAH [102], similar to BMPR2 and ALK1 mutations [103]. Caveolae act as membrane reservoirs to compensate for the increased membrane tension induced by flow, emphasizing their role in mechanosensing [104] (reviewed in [105]). Consequently, the caveolae flatten and disassemble [104]. Strikingly, the number of caveolae was reported to increase in response to physiological FSS in ECs [106,107]. This seemingly contradictory finding could provide an explanation to the dual role of caveolae in integrating FSS into the TGFβ/BMP signaling. First, an overall increase in FSS-induced caveolar endocytic turnover-rate might affect the TGFβ/BMP receptor degradation, recycling, or cytosolic retention. Second, flattening and disassembly of caveolae in response to FSS could impact on receptor oligomerization and SMAD activation. The particular shape of caveolae suggests that these structures prevent over-exposure of mechanosensors to shear stress, as already shown for VEGFR2. Since caveolae are found at the apical but moreover at the basal side of ECs, they provide islands for cooperative signaling between BMP receptors and integrins. Interestingly, it was reported that activation of integrin α5 by flow, requires its translocation to membrane lipid rafts [108], and that integrin mechano-transduction promotes cav-1 phosphorylation [109]. Furthermore, integrins seem to cluster in caveolae upon FSS [110], and caveolae-mediated integrin endocytosis regulates ECM remodeling [111]. Since integrin-BMP receptor interaction was suggested to participate in EC FSS sensation [112], caveolae could provide membrane environments for cooperative clustering. Interestingly, whereas cav-1 patches redistribute to the upstream edge facing the direction of flow, there is no change in the distribution of AP-1/2, which is indicative of CME [113]. As micropinocytosis, a process involving heavy cortical actin remodeling mediated by the small GTPase CDC42 [114], was described as the prevalent internalization route for VEGFR2, in the presence of ligand, this route could also be suggested for the TGFβ/BMP receptors in ECs. As with other analogies of TβRs/BMPRs to VEGFRs, one might raise the question whether these receptors are similarly affected by endocytosis. In addition to endocytosis, other cellular mechanisms were described to integrate mechanical forces into the endothelial TGFβ/BMP signaling. These mechanisms were associated with discrete subcellular compartments, which is discussed hereafter.

### 2.2. TGFβ/BMP-SMAD Signaling Crosstalk to Mechanobiology at Distinct Subcellular Compartments

In the following chapters, we walk through the distinct EC subcellular compartments proposed to incorporate cellular mechanics into the TGFβ/BMP signaling and vice versa (Figure 2). Given is an example of a muscularized artery (Figure 2a), with its different cell and ECM layers forming the vessel wall (Figure 2b). ECs interact with the ECM of the inner elastic membrane (IEM) and adjacent cells through FAs. These adjacent cells are collectively referred to as mural cells (e.g., pericytes and smooth muscle cells (SMCs)). Interaction of ECs to ECM (Figure 2b, I) allows for integrin-mediated ´outside-in´ or ´inside-out signaling´ [115]. Binding of ECM ligands to cell surface integrins stimulates conformational changes that induce intracellular signaling through integrin-associated proteins [116]. ´Inside-out signaling´ strengthens adhesive EC contacts and the appropriate force necessary for integrin-mediated cell migration, invasion, ECM remodeling, and traction force. Inside-out integrin-signaling allows cells to generate traction forces that participate in liberating latent ECM-bound TGFβ, for example. Since TGFβ/BMP signaling regulates the expression of a diverse set of integrins and ECM molecules, it directly modulates the availability of proteins required for such integrin-dependent cellular mechanics in a feed-forward fashion. We give examples of how FAs participate in the liberation of latent TGFβ and the localization of receptors, and how TGFβ/BMP signaling acts upstream of actin, ECM, and integrin regulation.

EC junctions (Figure 2b, II) resemble another subcellular compartment where crosstalk occurs. In a mature/quiescent blood vessel, ECs provide barrier function through tight lateral connectivity and a well-balanced equilibrium between cell-to-cell tension and cell-to-ECM traction. The equilibrium between these basolateral forces is important to maintain the barrier quality. We give examples of how TGFβ/BMP signaling is involved in regulation of this junctional force equilibrium.

One main mechanical force coupling to biochemical signals in ECs is the blood-flow generated FSS, which is sensed by the deflection of the primary cilium, for example (Figure 2b, III). Early reports on FSS-induced BMP-signaling suggest that FSS can activate TGFβ/BMP signaling, i.e., through receptor activation, which included integrin-mediated signaling [117] in the absence of TGFβ/BMP ligands. Such ligand-independent but FSS dependent activation was reported for VEGFR2, for example [118]. Recent work showed a requirement for at least a low concentration of endothelial BMP ligands for the effects of FFS on SMAD signaling, through careful titration and sequestration of BMP ligands from the tissue culture media [84,119]. The mode through which the primary cilium allowed for FSS-dependent TGFβ/BMP-signaling regulation is still not entirely understood. We give examples of FSS and cilia-related TGFβ/BMP research, and emphasize the role of TGFβ/BMP signaling as an upstream regulator of cilium formation.

Finally, we conclude our review by placing the EC nucleus into focus, together with the cell’s cytoskeleton as a mechanosensor (Figure 2b, IV). The nuclear compartment is linked via the cytoskeleton to the above-mentioned mechanisms, directing nuclear import of TGFβ/BMP signaling factors and changes in global chromatin architecture, conferring gene accessibility.

## 3. TGFβ/BMP Signaling Crosstalk at Focal Adhesions with Extracellular Matrix and Stiffness

There is a wealth of evidence that actin driven filopodia allow chemotactic and mechanical responses in ECs. The latter include durotaxis, the sensing of stiffness gradients at the tip of the cell, as well as tip-cell pulling forces (Figure 3a) that support the outgrowth of a new sprout (Figure 3b) [120,121]. Interestingly, VEGFR2 and its co-receptor neuropilin-1 accumulate at the distal filopodia tips [26,122,123]. Cytoskeletal rearrangements in proximity to BMP receptors are induced via ligand-induced non-SMAD pathways and rely on LIM kinase-cofilin [124], PI3K [125], and Rho GTPases [47] signaling. Additionally, we found that non-SMAD responses such as MAPK-p38-HSP27 signaling can induce EC migration [17] (Figure 3c).

ECM stiffness is beneficial for efficient sprouting angiogenesis in development and disease [126,127,128], and incorporates into the fate determination of different vascular beds [129,130]. As such, it was found that ECM stiffness controls lymphatic vessel formation through the regulation of GATA2, and subsequent downregulation of TGFβ2 on soft matrix [131]. Moreover, ECM stiffness is required for capillary maintenance and is related to vascular pathology [132]. Data on blood vessel stiffness largely differ depending on the method used, and can vary by several orders of magnitude [133,134,135], also depending on whether the whole vessel or only the subendothelial layers were analyzed [136,137]. It clearly differs between arterial and venous ECs, between species and between material of different age [138,139]. However, significant stiffness increase is found consistently in diseased endothelium, such as in atherosclerotic ApoE-null mice [140] or the fibrous cap in human carotid atherosclerotic plaques [141] and TGFβ-induced endothelial-to-mesenchymal transition (EndMT) of aortic valves [142], and it also occurs under higher matrix stiffness [143,144]. In arteries, increase in stiffness correlates positively with expression of BMP2 [145], as was reported for TGFβ2 in human umbilical vein ECs (HUVECs) [136].

Mechanical forces involved in sprouting angiogenesis include pulling by the tip-cell and possibly some pushing by the stalk-cells. It was observed through three-dimensional (3D) traction force microscopy that matrix deformations induced by tip-cells were dependent on actin-mediated force generation and correlated with sprout morphological dynamics. Furthermore, sprout tips were found to exert radial pulling forces on the ECM, possibly involving FAs [146]. Interestingly, high ECM stiffness increased the endocytosis rate of VEGFR2 and was involved in VEGFR2 clustering at the plasma membrane [147]. Cells in sparse culture exhibit less cell–cell contractility but the forces concentrate towards their substrate adhesion sites. A similar phenomenon for the TGFβ signaling was observed in fibroblasts. Here, ALK5 clustered to a higher degree with TGFβR2 at FAs, when cellular tension was reduced, which also increased the SMAD2/3 signaling. The same study suggested that FAs provide subcellular signaling hubs, in which the oligomerization and clustering of TGFβ receptors is regulated in a force-dependent manner [148] (Figure 3c).

### 3.1. Release of Latent TGFβ from Extracellular Depots by Mechanical Forces

Additionally to the regulation of receptor oligomerization, FAs provide a second layer of regulation, through which the mechanics are integrated into endothelial TGFβ/BMP signaling. Integrin-dependent traction forces emerging from FAs directly participate in liberating TGFβ as a biologically active growth factor, from its latent form (i.e., biologically inactive) (Figure 3c). In detail, TGFβ1/3 remains associated with a latency associated peptide (LAP) after secretion [149,150], which shields receptor-binding epitopes and preserves the inactive growth factor as part of an extracellular depot (Figure 3). Deposition of this latent TGFβ complex to ECM is facilitated via tethering to fibrillin fibers [151,152], through the adaptor protein latent TGFβ binding protein (LTBP) [153,154]. LTBP covalently links the latent TGFβ to the ECM, through an iso-peptide bond between LTBP and fibrillin [155]. Interestingly, it was found that activation of TGFβ ligands requires binding and pulling of specific integrins, such as αvβ1, αvβ6, and αvβ8 [156] on LAP RGD motif, to release the active forms of TGFβ1/3 [150,156,157] (Figure 3c). For this mechanism, it is important that the ECM, to which the latent complex is tethered, provides a certain degree of mechanical elasticity and thus provides resistance against the pulling forces applied by the cells. In fact, it was shown that ECM remodeling, preceding the activation step, mechanically primes latent TGFβ1, akin to loading a mechanical spring [158]. Hinz and co-workers found that stiff ECM allows a more efficient integrin-mediated TGFβ retrieval from the latent depot, compared to soft ECM [150] (Figure 3c). A soft matrix with an elastic modulus (*E)* ≤ 5 kPa preferentially deforms under the stresses applied by cells, leaving the latent complex intact and the TGFβ extracellularly sequestered. Whereas a stiff matrix with *E* >> 10 kPa resists deformation, resulting in distortion of the latent complex and the release of active TGFβ. The crucial role for ECM elasticity and integrin-mediated traction forces is underlined by the finding that absence of integrin-mediated TGFβ1 activation in vivo recapitulates the phenotype of TGFβ1-null mice [157]. Mechanistically, integrin binding to LAP allows conformational changes of the latent complex through actomyosin-generated tensile force, which act through the integrin β-subunit [158,159,160]. Structural analysis of latent TGFβ by the Springer group showed that this resembles the opening of a “straitjacket” [161]. Interestingly, TGFβ released by such mechano-dependent mechanism signals in close vicinity to the cell pulling [162], creating a local niche environment enriched in TGFβ. Forces required to induce a conformational change in LAP were measured to be in the tens of piconewton range [160]. It is to be noted that this level of force can be sustained by individual integrins and produced by few myosin motors, and it is a thousand times lower than the forces transmitted by one single FA [163]. Mature BMPs were also shown to bind to ECM, including tenascin-c, heparins, and laminins [164,165,166], creating an extracellular BMP growth factor depot. In contrast to integrin-mediated pulling forces, their bioavailability might be dependent on mechanisms including secretion of matrix metalloproteinases (MMPs), leading to ECM degradation and subsequent BMP release (Figure 3c). Here, MMP expression in ECs was shown to be dependent on cyclic stretch and strain [167,168,169]. TGFβ also acts upstream of integrin and ECM expression (Figure 3c), highlighted by findings that TGFβ2 stimulation of ECs can directly enhance cell-matrix traction stresses [136]. TGFβ induces expression of integrins α2, α5, αv, β1, and β3 in both microvascular [170] and macrovascular ECs [171,172]. Moreover, TGFβ induces expression of ECM proteins, including fibronectin and collagens I, IV, V (Figure 3c). We showed previously that depletion of endothelial BMPR2 shifts signaling responses from BMP towards TGFβ through increased mechanical retrieval of active TGFβ from the ECM. BMPR2 deficiency led to enhanced contractile forces and stiffness of ECs [173]. This suggests that unbalancing BMP and TGFβ signaling is sufficient to induce major mechanical alterations on the level of cell adhesion and composition of the ECM. Moreover, TGFβ/BMP receptors might directly cooperate with integrins. It was shown before that endoglin mediates fibronectin/α5β1 integrin and TGFβ pathway crosstalk in ECs [174]. Fibronectin and its cellular receptor α5β1 integrin specifically increase TGFβ1- and BMP9-induced Smad1/5/8 phosphorylation, via endoglin and ALK1. Endoglin controls cell migration and composition of FAs by interacting with the FA-related protein zyxin [2]. Upon BMP9 stimulation, endoglin regulates subcellular localization of zyxin in FAs [175]. Latent TGFβ becomes biologically active, not only by cell-mediated integrin pulling, but also by the shear forces acting on the latent complex [176,177]. More recently, it was found that oscillatory shear stress (OSS) potentiates latent TGFβ1 activation, more than laminar FSS. Abrupt changes in rotation direction seem sufficient to recruit mature TGFβ from the latent complex [178]. These findings highlight the role of excessive TGFβ signaling for vascular pathologies that involve dysregulated integrin expression on one hand, and disturbed flow regimes on the other. Together, at the FA compartment, integrin-mediated traction/pulling forces on the ECM or latent TGFβ provides a delicate mechanism of how ECs mechanically interact with the environment. Interfering with such mechanisms or the TGFβ/BMP-signaling dependent upregulation of integrins or ECM proteins could be a promising strategy to target vascular pathologies.

## 4. BMP/TGFβ Signaling and Integration of Basolateral Forces

Mature ECs are mechanically coupled both to the ECM and via cell junctions to neighboring cells (Figure 4). However, the balance, coordination, and interdependency of these forces are poorly understood. Junctional resolution is a hallmark of EndMT, characterized by loss of EC junctional proteins such as vascular endothelial (VE)-cadherin and Platelet and Endothelial Cell Adhesion Molecule (PECAM)-1, both essential to maintain cell barrier integrity. EndMT is favored under perturbed BMP and TGFβ signaling in disease, but also in normal development, such as formation of the endocardial cushions in the atrioventricular canal [62,179], and is characterized by the expression of *Snail*, *Slug*, and *Twist* genes [179,180,181].

It was recently found that force sustained at the cell–cell contact between epithelial cells is approximately 100 nN, directed perpendicular to the cell–cell interface and concentrated at the contact edges [182]. Intriguingly, a direct relationship between the total cellular traction force on the ECM and the endogenous cell–cell force exists, indicating that cell–cell tension is a constant fraction of cell–ECM traction (Figure 4a) [182]. Mechanisms disrupting this balance, e.g., up-regulation of cell-substrate adhesion-related proteins such as integrins, directly imbalance EC cell–cell connectivity and thus the EC barrier function. While cell–ECM traction is facilitated via ECM-bound integrin clusters at FAs, cell–cell tension is mediated via cell junction proteins like VE-cadherin and PECAM-1, providing mechanical stability by zipper-like trans-intermolecular interactions (Figure 4). If seen as force vectors, force distribution is thus balanced between apico-basal traction forces and tensile forces in the lateral plane, or in other words, cell adhesion forces influence junctional forces (see Figure 4b,c). EC junctions can be subdivided into tight (TJs), gap (GJ), and adherens junctions (AJs) [183]. TGFβ/BMP signaling integrates into balancing junctional forces twofold. First, TGFβ/BMP signaling regulates integrin expression and ECM deposition, as mentioned earlier, impacting the cell–ECM traction, and second, lateral cell–cell tension is affected by the TGFβ/BMP-dependent regulation of junctional protein localization, expression, and their turnover/endocytosis (Figure 4).

We recently showed that angiomotin (AMOT), a TJ-associated membrane protein, drives BMP-SMAD signaling at the apical membrane in polarized cells. In HUVECs, AMOT interacts with BMPR2 and knockdown of AMOT reduces SMAD signaling, only from the apical side of polarized cells, while basolateral BMP-SMAD signaling remained unaffected [4] (Figure 4a, upper). For example, microvascular beds specifically rely on TJs, as part of the blood–brain barrier (BBB) [184]. TGFβ1 was shown to promote barrier function upon maturation of corneal ECs [185] and to upregulate tight junction- and P-glyco-proteins in brain microvascular ECs [186]. Further, depletion of TGFβ1 in vivo and in vitro leads to a breakdown of the microvascular blood–retinal-barrier. This was characterized by decreased association between the TJ proteins ZO-1 and occludin [187]. Second, concerning GJs, it was demonstrated that Cx37 (Connexin37) is a differentially regulated target of ligand-induced and mechano-transduced SMAD1/5 signaling, and that the Cx37-loss enables pathological vessel enlargement and shunting [119]. Moreover, the ALK1 signaling axis was found to regulate endothelial Cx40 expression [188]. AJs contribute significantly to the EC barrier function and are prone to regulation by FSS, which reorganizes junction-associated proteins [189]. We recently reported that BMP6 leads to Src-dependent phosphorylation of VE-Cadherin and subsequent internalization (Figure 4a, depicted in center), ultimately leading to EC hyperpermeability [29]. Concordantly, BMP4 was reported to regulate leukocyte diapedesis and promote EC inflammation, while BMP4 KO leads to increased VE-Cadherin expression [190]. Both BMP2 [191] and BMP4 increased vascular permeability [192]. These results in summary showed that BMP2/4/6 induce EC permeability, likely resulting in reduced cell–cell tension. In contrast, BMP9/ALK1 signaling prevents vascular permeability and was recently shown to prevent the leakiness of the inflamed lung microvasculature [193]. BMP9-ALK1 signaling facilitates cell–cell contacts and balances the interdependency between apico-basal and lateral forces [194]; Figure 4b,c. This is further supported by the finding that BMP9/ALK1 signaling prevents VEGF-induced phosphorylation of VE-cadherin, and induces the expression of occludin, thus, strengthening the vascular barrier functions [194]. Third, the VE–cadherin–catenin complex forms the molecular basis of AJs and cooperates with actin filaments, controlled by the Arp2/3 complex. While fully mature EC monolayers have long and tight junctions, activated ECs display more transient and lesser stable cell-to-cell interactions. The junctional dynamics through which ECs form these highly dynamic interactions were studied in detail in the presence of FSS [195], and were termed junction-associated intermittent lamellipodia (JAILs) [196]. JAILs allow for subcellular organization of junctional forces homing clusters of VE-Cadherins [196]. Dynamic JAIL remodeling of confluent EC monolayers exposed to FSS is important to maintain barrier integrity or allow for cell migration and angiogenesis [197]. AJs are rich in a FSS mechanosensitive super-complex, composed of VE-Cadherin, VEGFR2/3, and PECAM1 [198,199]; Figure 4a, center. However, it is not clear whether this complex acts in cooperation with or independent of FSS-sensitive mechano-complexes, comprising integrins and BMP receptors [112]. In support of such cooperation is that VE-cadherin was reported to co-immunoprecipitate with all components of the TGFβ receptor complex—TGFβRII, ALK1, ALK5, and endoglin. Accordingly, ECs lacking VE-cadherin are less responsive to TGFβ/ALK1- and TGFβ/ALK5-induced SMAD phosphorylation and target gene transcription [200]. Interestingly, AMOT can retain YAP/TAZ in the cytosol and VE-Cadherin tethers it to stable junctions [201,202,203]; Figure 4a. In line with this, excluding endothelial YAP/TAZ from the nucleus by tethering it to junctions, i.e., through mechanical regulation, such as reduced junctional tension, could limit nuclear SMAD signaling, as shown for fibroblasts, for example [73]. However, Gerhardt et al. recently found that YAP/TAZ act upstream of junctional VE-cadherin turnover and that nuclear YAP and TAZ also decreases endothelial BMP signaling, possibly by increasing the expression of BMP inhibitors [204]. Together, it could be concluded that EC junctions build a discrete subcellular compartment, where TGFβ/BMP signaling integrates into mechanobiology, and that in healthy blood vessels, a balanced equilibrium of basolateral forces is tightly coupled to a similarly balanced TGFβ/BMP signaling. Targeting this in the context of pathological changes in vascular barrier functionality would be an interesting opportunity.

## 5. Shear-Stress-Induced TGFβ/BMP Signaling at the Primary Cilium

EC phenotype and homeostasis are maintained by a unidirectional, laminar blood flow, ranging approximately from 1 to 5 Pa (Pa; equals 10–50 dyn/cm^2^) in human vessels [205]. This laminar fluid shear stress (FSS) is typically vaso/aterio-protective. The vessel architecture at bifurcations, curvatures, and rough luminal topographies, could create inhomogeneous flow dynamics, such as at venous and lymphatic valves, termed oscillatory shear stress (OSS) or disturbed shear stress (DSS). OSS and DSS are characteristic of low shear stress, flow reversal, and in the latter case, chaotic turbulent flow regimes. Regions of OSS/DSS blood flow perturbation, together with hyperlipidemia, could show signs of inflammation, including low nitric oxide production, reduced barrier function, and increased pro-adhesive, pro-coagulant, and pro-proliferative properties. Shear stress is sensed mainly by cells, through the primary cilium (Figure 5a), a single microtubule-driven membrane protrusion on the apical side. However, according to the decentralized model of endothelial mechano-transduction [206], FSS integrates into all compartments mentioned in this review and a single shear stress mechano-transducer is unlikely to exist.

The main flow-induced force acting on the ciliary membrane is bending vs. relaxation [207] (Figure 5a). Cilia deflect sideward, in response to unidirectional low shear stress (LSS) or high shear stress (HSS) [208], whereas they deflect in alternate directions at OSS or DSS, according to a heavy elastic model [209] (Figure 5b). Consequently, a direct activation of ciliary-membrane-located receptors involves conformational changes induced by cilia mechanical bending [209,210], as best studied for calcium channels [211], depicted in Figure 5. Interestingly, depletion of osmolarity and cell volume regulating calcium channel TRPV4 in mice, completely abolishes shear stress induced vasodilation [212]. TRPV4 localizes to mesenchymal stem cell cilia [213] and it was shown that the differentiation process of cardiac fibroblasts to myofibroblasts is dependent on TRPV4, which integrates mechanical (i.e., ECM stiffness) and TGFβ signals [214]. Moreover, interfering with TRPV4 signaling blocks TGFβ-induced epithelial-to-mesenchymal transition (EMT) in normal mouse primary epidermal keratinocytes (NMEKs) [215]. It is thus tempting to speculate that TRPV4 might also be a regulator of mechano-TGFβ crosstalk in the endothelium.

Receptors located at the tip of the cilium are activated upon ciliary deflection [213]. TGFβ/BMP signaling was shown to regulate the formation and length of primary cilia [216,217], including regulation of IFT88 expression [218], a major ciliary structural protein. In ECs, a lack of primary cilia primes shear-induced EndMT via the TGFβ-ALK5-SMAD2/3 axis [219] and ECs lacking primary cilia are sensitized to undergo TGFβ/BMP-induced EndMT (reviewed in [181]). This is in line with the finding that *Slug* is expressed in ECs lacking primary cilia, promoting EndMT and cellular calcification [220]. Furthermore, TGF-β receptors were shown to be endocytosed at the pocket region of the primary cilium (Figure 5), while TGFβ stimulation increased receptor localization and activation of SMAD2/3 and ERK1/2 at the ciliary base, along with SMAD4 accumulation [221] (Figure 5a).

With regards to endothelial ALK1-BMP9 signaling, primary cilia sensitize ECs to BMP9 ligands and prevent excessive vascular regression [222]. Here, it was proposed that BMP9 signaling positively cooperates with the primary cilia at low flow, to keep the immature vessels open before high–shear-stress-mediated remodeling [222]. The finding that the ALK1-BMP9-SMAD1/5/9 signaling axis is specifically enriched at the primary cilia, is in line with the observation that ALK1 appears enriched around the cilium. Moreover, prominent phosphorylation of SMAD1/5/9 was found at the basal body and along the cilium. Furthermore, FSS increases the physical interaction between ALK1 and endoglin, and thereby lowers the effective concentration of BMP9 required for ALK1 activation [84]. FSS-enhanced association of Alk1 and endoglin, showed that Alk1 is required for BMP9 and flow responses, whereas endoglin is only required to enhance the effects of flow [84]. Finally, both TGFβ and BMP signaling is fine-tuned in a ciliary-compartment-specific manner through several PTMs at the receptor- and SMAD-levels. For example, it was found that the E3 ubiquitin ligase SMURF1 localizes to the cilium and controls local BMP signaling [223] (Figure 5a).

### The Endothelial Glycocalyx

A significant portion of the apically applied FSS is absorbed by the glycocalyx, a thick coat of glycoproteins and proteoglycans that is negatively charged, forming a brush-like cushion on the surface of ECs (Figure 5a, upper). The glycocalyx varies in its viscoelastic properties, due to the degree of crosslinked side chains [224] and has a thickness reaching up to 0.5–3 μm into the lumen of vessels. This relatively large glycocalyx exceeds the thickness of the endothelium and the length of several apically localized transmembrane receptors, including leucocyte adhesion molecules [225] and TGFβ/BMP receptors, if not localized to ciliary tips surpassing the glycocalyx. The glycocalyx was found to be thinner in atherosclerotic risk regions, where OSS/DSS can be expected [226], while it is thickened under protective FSS [227]. Moreover, proteases involved in ECM degradation, and consequently releasing ECM-bound growth factors, are expressed upon FSS [228,229], as well as furin, a pro-TGFβ processing protease [230]. Since both TGFβs and BMPs are positively charged and were found to bind to negatively charged heparins [231,232], it was proposed that the glycocalyx could provide a superb reservoir for TGFβ/BMP growth factors, as well as their antagonists, including Noggin [233,234].

Together, the primary cilium acts as a small but important center for integrating FSS into TGFβ/BMP signaling. Current advances in super resolution microscopy and development of in vitro flow devices will add further knowledge on how TGFβ/BMP receptors and their signaling are regulated at the primary cilium. The modes of how FSS and the glycocalyx regulate TGFβ/BMP signaling and bioavailability are important to understand vascular diseases, e.g., arteriosclerosis.

## 6. Integrating Forces into Nuclear TGFβ/BMP Signaling

The term shear stress response element (SSRE) was first proposed after the identification of the sequence GAGACC, within the shear stress responsive gene promoters [235], and was later extended to a number of other shear stress responsive elements. Among several *cis*-acting elements that were implicated in mediating shear modulation of gene expression, one example of a factor that binds to the GAGACC promoter region, is the transcription factor nuclear factor κ-B (NF-κB) p50–p65 heterodimers. Another factor found to bind SSRE is myocyte enhancer factor 2 (MEF2), downstream of MAPK or PI3K signals, which regulates the well-studied shear–stress-induced expression of Krüppel-like factors (KLFs) [236]. One mechanism making shear–stress-sensitive genes competent for transcription, is the regulation of the nuclear architecture itself. Here, the EC nucleus is appreciated as a direct sensor of blood flow direction [237] and possesses its own mechanosensitive apparatus [238]. Global EC responses to FSS include nuclear elongation, position adaptation of nuclear-envelope-associated organelles against the direction of flow [239] and a ´streamlined´ cell morphology [240]. When ECs adapt to shear stress, forming an upstream edge, the centrosome positions relative to the nucleus, driven by the actin and microtubule networks in a ´string puppet´-like fashion (Figure 6a,b). Indeed, the centrosome is an ideal integrator of extracellular and intracellular mechanical signals [241]. Interestingly, ECs with depletion of SMAD4 fail to migrate against the direction of FSS [16].

### 6.1. Forces Acting at the Nuclear Envelope

Nuclear import of several transcription factors is altered upon the application of FSS, including YAP [242]. One mode of how forces promote nuclear import of transcription factors is direct force transmission towards stretching and widening of nuclear pore complexes (NPCs), as shown for YAP shuttling [243]. Impressively, mechano-transduction bypasses the kinetics of biochemical signaling, as shown for Src (<0.3 s of mechanical Src activation vs. >12 s by soluble epidermal growth factor stimulation) [244]. Recent reports indicate that forces are directly transmitted from the ECM to the nucleus, via the cytoskeleton (reviewed in [238,245]). Physically, forces are coupled to the nucleus via a mechanical hardwiring established between the junctional cadherins and integrins within FAs that enable bridging of cytoskeletal filaments, which reach out to the linker proteins attached to the nuclear envelope [246]; Figure 6b,c. The actin and microtubule cytoskeletons exert differential control on nuclear morphology [247]. While the actin and actomyosin fibers provide tensile forces to the nucleus in a lateral plane, mechanical loading of the nucleus is facilitated by compression through microtubules on top of the nucleus [247,248,249,250] (Figure 6a,b). Furthermore, cells can protect their nucleus from deformation through an actin cap, shielding the nucleus from mechanical stress from the top [251]. A proper positioning of the actin cap is dependent on basal zyxin FAs, with perinuclear localization [252,253]. Concerning the nuclear structure, the nuclear envelope is composed of the outer nuclear membrane (ONM), and the inner nuclear membrane (INM), separated by the perinuclear space (PNS) (Figure 6c). A key feature of the nuclear envelope is the nuclear lamina (NL), a polymer mesh lining the inner surface of the INM, formed by lamins. These levels of lamin A/C determine the nuclear envelope stiffness and deformability [254,255,256]. Supporting this, lamin A/C-deficient cells exhibited increased numbers of misshaped nuclei and severely reduced nuclear stiffness [255,257]. Interestingly, ECM stiffness directly influences lamin A expression [258], and LSS (5 dyn/cm^2^) leads to decreased lamin A/C expression in ECs [259]. However, this differs between the young and aged ECs [260]. Regarding TGFβ signaling, a deficiency in A-type lamins correlates with the hyperactivation of TGFβ signaling [261,262]. In epithelial cells, TGFβ-induced EMT leads to downregulation of lamin A/C, lamin B, the nuclear envelope membrane protein emerin, and multiple nucleoporins forming the NPC [263,264]. Further crucial nuclear envelope proteins are integral inner nuclear membrane proteins, such as MAN-1. MAN-1 binds R-SMADs but not co- and I-SMADs, thereby competing with the transcriptional co-factors for binding to SMADs (Figure 6c). Additionally, MAN-1 facilitates SMAD dephosphorylation through phosphatase PPM1, which represses signaling by both TGFβ and BMPs [265,266,267,268,269]. Mechanical force transduction acting from outside of the cell towards the nuclear envelope requires tethering of cytoskeletal elements to the Linker of Nucleoskeleton and Cytoskeleton (LINC) complex, formed by the interaction of nuclear envelope spectrin-repeat proteins (nesprins) and Sad1p, UNC-84 (SUN) proteins [245,270,271]. Both nesprin and SUN proteins are downregulated in ECs, when exposed to LSS, providing a relevance for altered nuclear mechano-sensation via LINC, during vascular dysfunction [272]. Different nesprin isoforms orchestrate coupling of the SUN complex to either actin fibers, microtubules, or intermediate filaments (Figure 6c). Importantly, there is strong evidence that nesprins also influence SMAD nuclear shuttling, as depletion of nesprin-2 in HaCaT cells lead to slower TGFβ-induced nuclear translocation of SMAD2/3. Here, nesprin, based on its ability to interact with emerin and lamin, was suggested to affect TGFβ/SMAD signaling, as emerin is directly involved in this signaling pathway through its interaction with MAN-1 [273].

### 6.2. Forces Acting on the Chromosomal Architecture

Cytoskeletal forces acting on the LINC complex influence heterochromatin localization and core histone protein mobility, and thereby, directly alter gene transcription and induce epigenetic changes (Figure 6d). Transcriptionally inactive heterochromatin, which is marked by a distinct pattern of histone trimethylation and acetylation, tends to localize to the NL, creating the so-called lamina associated domains (LADs) [274,275] (Figure 6d). Interestingly, mature ECs with a quiescent phenotype have a high dependency on epigenetic regulators that maintain this phenotype. Transcriptomic and epigenetic mapping of quiescent lung endothelium revealed that inhibitory SMAD6 and SMAD7 are particularly epigenetically regulated in quiescent ECs, to control TGFβ signaling [276]. SMADs also co-immunoprecipitate with histone-modifying enzymes (HME, Figure 6c) such as histone deacetylases (HDACs), homeodomain protein TG-interacting factor (TGIF), and histone methyltransferases. This provides a mechanism through which SMAD binding to the DNA influences the epigenetic landscape in vascular development and homeostasis [277,278], and provides a mechanism that allows SMADs to become transcriptional repressors [279,280].

### 6.3. Forces and SMAD Transcriptional Co-Factors

A third way through which mechanics integrate into nuclear TGFβ/BMP signaling is through the SMAD dependency on transcriptional co-factors that act in *cis* or *trans* to recruit additional enhancers to the SMAD-binding elements. Thus, they increase SMADs transcriptional activity and direct the SMAD complex to particular target genes (Figure 6). Two mechanically regulated pathways were shown to integrate into SMAD signaling in ECs—YAP/TAZ and myocardin-related transcription factor-A (MRTF-A) signaling. TGFβ was shown to induce formation of YAP/TAZ–Smad2/3-4 complexes [71,281] and BMPs were shown to induce YAP/TAZ-Smad1/5-4 complexes [282,283], however, there is evidence that SMADs 2/3 and SMADs 1/5/9 react differently to YAP/TAZ incorporation, with respect to their transcriptional activity. YAP/TAZ are incapable of directly binding DNA, but instead they interact with the TEA domain transcription factors (TEADs) [284]. Mechanistically, YAP/TAZ in complex with TGFβ-SMADs (SMAD2/3) and co-Smad4, require concomitant binding to SMAD binding elements (SBEs) and TEAD promoter elements, such as that found in promoters of cysteine-rich angiogenic inducer 61 (*CYR61*) [285,286] and connective tissue growth factor (*CTGF*) [287]. For BMP SMADs, it was found that Smad1/5/Smad4 binding with YAP/TAZ competed with YAP for the TEAD interaction and inhibited YAP’s co-transcriptional activity [283]. However, data also showed that BMP-SMAD1/5 interaction with YAP/TAZ increased transcriptional activity, which might depend on a stabilizing role of YAP binding for SMAD protein [282,288].

A delicate mechanism through which MRTF-A transcription factor integrates into the mechano-dependency of TGFβ/BMP SMAD signaling, is the ability of MRTF-A to directly sense the status of actin remodeling, by binding to globular actin (G-actin) (Figure 6c). MRTF-A is required for TGF-β2-induced α-SMA expression [289] and regulates *Slug* expression in synergy with TGFβ [290]. MRTFs are co-activators of serum response factor (SRF)-dependent transcription [291] and MRTF-A was shown to interact with TGFβ-SMAD3 [290] and BMP-Smad1 [292], to co-activate the expression of SMAD target genes (Figure 6). In the repressed state of MRTF-A, monomeric G-actin binds in a 5:1 complex to MRTF-A. Transcriptionally inactive G-actin:MRTF-A complexes are found both outside and inside the nucleus, and either inhibit the nuclear import or foster the nuclear export of MRTF-A [293]. Importantly, MRTF nuclear translocation and transcriptional activation depends on competition with actin regulatory proteins, such as the Wiskott–Aldrich syndrome protein (WASP) for G-actin binding [294]. In TGFβ-induced EMT, nuclear localization of MRTF-A correlates directly with mechanical stress, tissue geometry, and the resultant variability in cytoskeleton dynamics of epithelial cells [295]. Mechanistically, the same scenario is likely to occur during TGFβ-induced EndMT. Furthermore, the SMAD-MRTF-A complex could form an interesting Ménage-à-trois relationship with the aforementioned YAP/TAZ transcription factors. While functional relationship between YAP/TAZ and MRTF-A in ECs was suggested before [296], it could be speculated that cooperation of this complex exists together with SMADs [5,297] (Figure 6c).

## 7. Perspective and Concluding Remarks

As summarized here, crosstalk of endothelial mechanobiology and TGFβ/BMP signaling has a pivotal role for endothelial physiology and pathology. The various mechanisms through which mechanobiology integrate into this delicate signaling network create another layer of regulation that participates in fine-tuning and diversifying TGFβ/BMP signaling. We and others aim to understand the precise molecular mechanisms behind this signaling crosstalk. Such basic molecular investigations remain challenging in an in vivo setting. While previous data gained under static EC culture are a first approximation, technical developments for in vitro tissue culture now allow us to study ECs signaling under biochemical and mechanical conditions, which make these data more relevant to the human system.

We have cited examples of how TGFβ/BMP signaling integrates into mechano-transduction in a compartment-specific manner. With crosstalks at discrete membrane domains, the cytoskeleton, the primary cilium, and the nucleus listed, show some of the most crucial compartments integrating EC mechanics into TGFβ/BMP signaling and vice versa. Since investigating endothelial TGFβ/BMP mechano-crosstalk is a relatively young research field, we cited a number of research articles on non-endothelial cells, where we considered as a wealth of evidence that in ECs it might be mechanistically similarly relevant. However, it remains to be proven whether all of the mentioned crosstalk mechanisms were conserved between other cell types and ECs. Further research will help answer these questions and to better understand crosstalk mechanisms of TGFβ/BMP signaling with mechanobiology that underlie human vascular pathologies.

## Figures and Tables

**Figure 1 cells-09-01965-f001:**
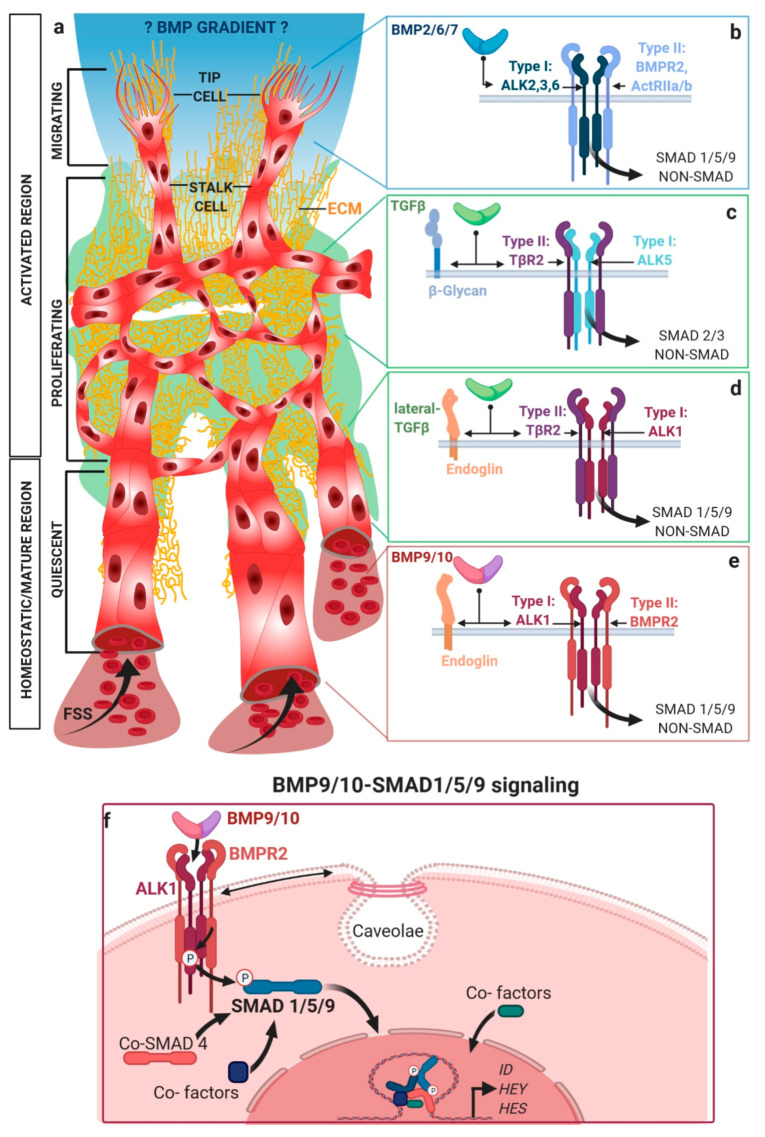
Different BMPs/TGFβ ligands induce activating or quiescent/homeostatic functions on endothelial cells (ECs). Depicted is the outgrowth (upper), pruning (middle), and maturation (lower) of a developing vascular network, (**a**) with an activated region (upper/middle) and a more quiescent, homeostatic region (lower). Characteristic at the active angiogenic front (upper) is the induction of sprouting angiogenesis, with distinct tip-cells at the leading front that utilize filopodia to sense- and pull- the extracellular matrix (ECM), followed by the proliferating stalk cells. Angiogenic gradients of activating BMP2, 6, or 7 (blue) are proposed based on in vitro data. TGFβ (green) bio-availability is tightly controlled through the interactions of the latent TGFβ complexes, with the underlying ECM (yellow). Intraluminal blood flow (lower) internally provides systemic BMPs (such as BMP9/10 (purple)) to ECs, inducing their maturation and ultimately maintaining a quiescent endothelial phenotype. Hemodynamics of blood flow exert concomitantly mechanical inputs through generation of fluid shear stress (FSS). The corresponding TGFβ/BMP receptor complexes and signaling branches proposed to be involved in these processes are shown on the right. Activating BMP2, 6, or 7 (blue) signal via R1s ALK 2, 3, and 6, together with R2 BMPR2 or ActRIIa/b to induce SMAD1/5/9 signaling or non-canonical responses (**b**). TGFβ (green) signals via ALK5 and TβR2 to activate SMAD2/3, (**c**) while it can also signal via ALK1 and TβR2, to activate SMAD1/5/9 signaling (lateral signaling) (**d**). Co-receptors like Endoglin and beta-glycan help presenting ligands to the receptors on the surface of ECs. BMP9/10 signaling via ALK1 and BMPR2 provides quiescence signals (**e**). Nuclear SMAD translocation and co-factor incorporation is shown in (**f**) at the example of BMP9/10-mediated SMAD1/5/9 signaling, inducing the transcription of EC genes. Among them are the inhibitor-of-differentiation (*ID*), *HES*, and *HEY* genes. ALK1 is located in caveolae, membrane structures that are mechanically regulated by FSS, for example.

**Figure 2 cells-09-01965-f002:**
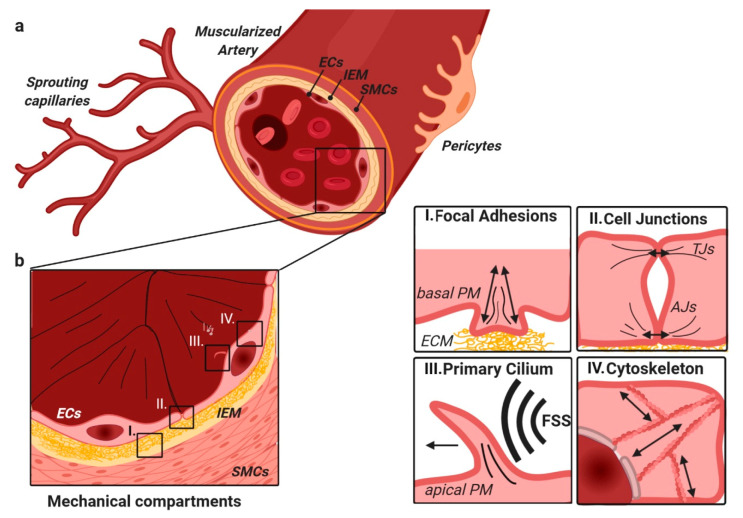
Muscularized arteries are composed of different layers of cells and interconnective ECM (**a**); from inside to outside—ECs, the IEM, SMCs, and pericytes (**b**). In ECs, mechanical signaling is found at four major hubs (enlarged on the right): I. Focal adhesion-ECM contact sites, important in cellular migration, and adhesion. II. Cell–cell junctions that are highly important in the regulation of EC barrier function and cell–cell communication. This includes AJs, TJs, and gap junctions that contribute differently. III. The primary cilium, which is apically located in ECs and initiates signaling upon deflection by FSS, for example. IV. The cytoskeleton that links the plasma membrane with the nucleus, thereby allowing direct force transmission. Abbreviations: ECs—endothelial cells, IEM—inner elastic membrane, SMCs—smooth muscle cells, FSS—fluid shear stress, AJs—adherens junctions, TJs—tight junctions, and PM—plasma membrane.

**Figure 3 cells-09-01965-f003:**
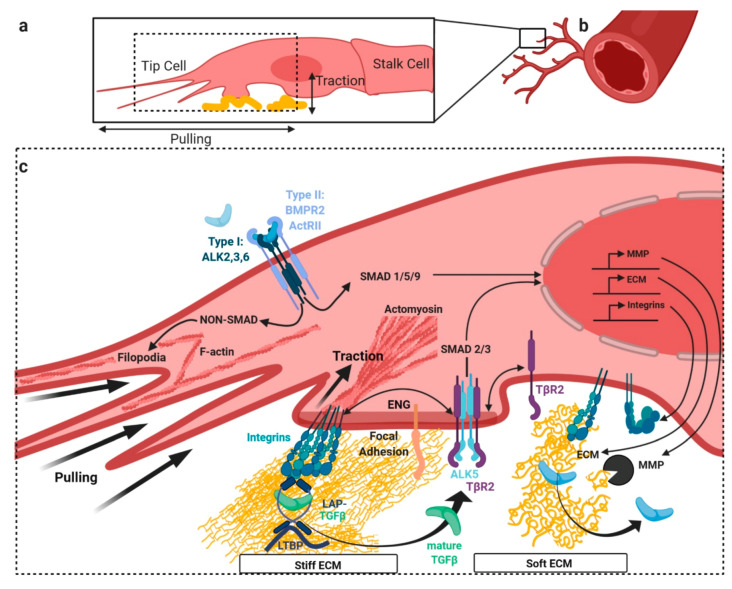
Crosstalk with the TGFβ/BMP SMAD pathway at focal adhesions (FA). In angiogenesis tip-cells at the sprouting front (**a**) emerging from pre-existing blood vessels (**b**) display enhanced FA-ECM contacts and filamentous (F-) actin driven filopodia (**c**). Exemplified at the sprouting front (lower caption), actin reorganization is in part orchestrated by non-SMAD signaling. FAs influence mechanical properties of the ECM, including stiffness. These contacts are mainly mediated by integrins, providing the cell with inside-out signaling properties. Integrins can directly bind ECM molecules outside the cell and are indirectly bound to the intracellular contractile actomyosin cytoskeleton. Endoglin (ENG) interacts with both, integrins and signaling receptors. TGFβ in its latent form (LAP-TGFβ) is bound to ECM filaments via LTBP. Viscoelastic properties of ECM directly influence integrin-mediated TGFβ retrieval from LAP-TGFβ-LTBP complexes. Integrins release mature TGFβ from the latent complex by exerting pulling forces, through the contraction of the actomyosin cytoskeleton. Stiff ECM allows more efficient release of mature ligand than soft ECM. In a feedback-loop, SMAD signaling in turn regulates the expression of ECM genes, integrins, and MMPs, which support the softening and the release of ligands by degrading the ECM. Abbreviations: ENG—endoglin, LAP—latency associated peptide, ECM—extracellular matrix, LTBP—latent TGFβ- binding protein, and MMPs—matrix metalloproteinases.

**Figure 4 cells-09-01965-f004:**
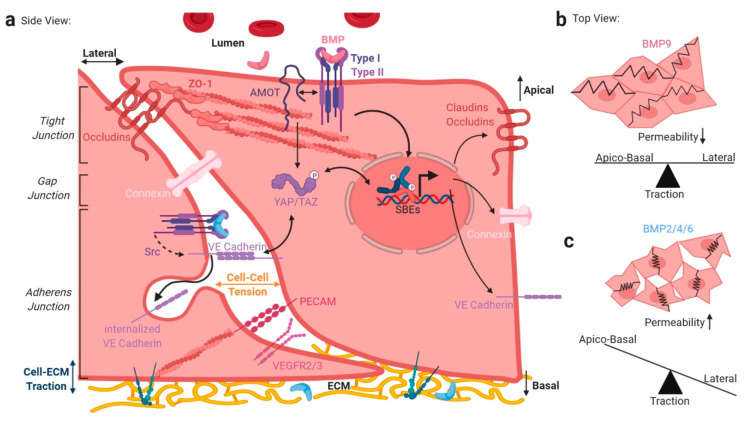
Crosstalk of mechanical and TGFβ/BMP signaling at endothelial cell junctions. EC junctions are crucial for cell–to-cell communications and barrier function (**a**). Apical TJs, composed mainly of claudins and occludins connect to the cytoskeleton via zona occludens (ZO) proteins, regulating trans-endothelial macromolecule transport by narrowing the inter-endothelial junction size. AJ-related protein AMOT regulates the apical BMP signaling by interaction with BMP receptors. Gap junctions allow communication between neighboring ECs via pore-forming proteins like connexins. Basal AJs constitute connection points of cell–ECM and cell–cell interactions. The main components are the trans-interacting proteins VE-Cadherin and PECAM-1, together with VEGFR2/3, which form a mechanosensory complex, necessary for FSS sensation. VE-Cadherin internalization and thus AJ resolution is regulated by BMP signaling via Src. Additionally, VE-Cadherin associated β-Catenin complex interacts with YAP/TAZ in the cytoplasm, and YAP/TAZ interacts with SMADs proposed to influence SMAD stability, shuttling, and transcriptional competence. SMAD gene transcription regulates the expression of several junctional proteins like claudins, occludins, connexins, or VE-Cadherin. Here, distinct BMP ligands exhibit different potentials of junction regulation and thus cell–cell and cell–ECM traction (**a**). As proposed on the right, homeostatic BMP9 might decrease EC permeability by keeping a balance of apical–basal and lateral traction (**b**). In contrast, BMP2/4/6 might increase EC permeability by shifting towards apico-basal traction (**c**). Abbreviations: SBE-—MAD binding elements, TJs—Tight junctions, AJs—Adherens junctions.

**Figure 5 cells-09-01965-f005:**
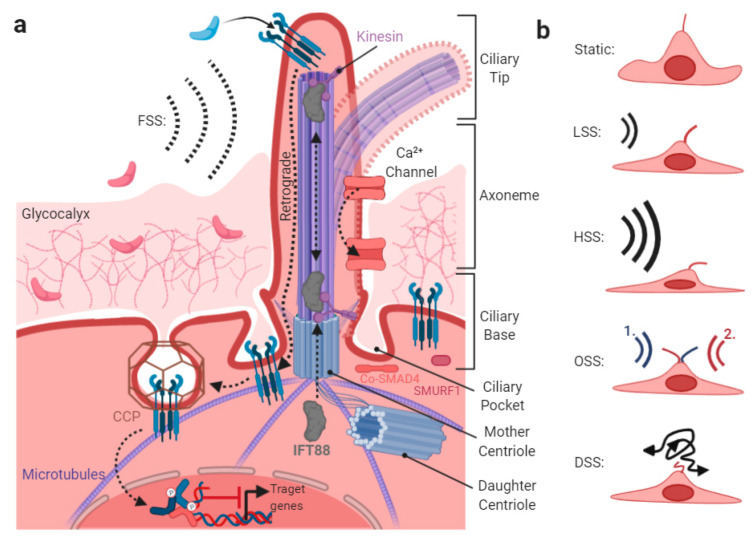
Crosstalk of fluid shear stress and TGFβ/BMP signaling at the primary cilium. The primary cilium is composed of a mother and a daughter centriole connected to other cellular compartments (like the Golgi apparatus), through extended microtubules. The axoneme, consisting of 9 pairs of microtubules, extends from the mother centriole in the vascular lumen, protrudes the PM, and exceeds the surrounding glycocalyx. Build up and maintenance of the axoneme is regulated by the shuttling proteins, e.g., IFT88 connected to motor proteins like kinesin. BMP/TGFβ receptors are mainly found at the ciliary tip, the most distal part of the axoneme. Upon ligand binding, retrograde shuttling of receptors towards the ciliary base, the part proximal to the mother centriole, is initiated. The ciliary base is a hotspot for clathrin-mediated endocytosis, which might be crucial for tuning cellular sensitivity to TGFβ, by regulating receptor endocytosis. Additionally, SMAD4 and SMURF1 are found at the ciliary base (**a**). Deflection properties of cilia in response to different FSS regimes are indicated on the right (**b**). Abbreviations: CCP—clathrin-coated pit, FSS—fluid shear stress, LSS—low shear stress, HSS—high shear stress, OSS—oscillatory shear stress, and DSS—disturbed shear stress.

**Figure 6 cells-09-01965-f006:**
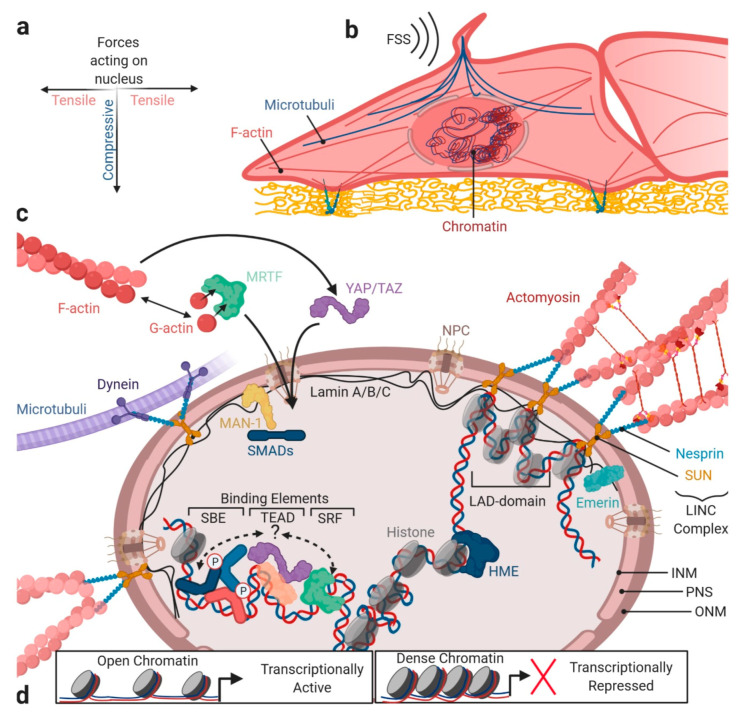
TGFβ/BMP signaling crosstalk and nuclear mechanics. The nucleus is connected to the PM, including the primary cilium, junctions, and cell–ECM contacts via the F-actin fibers and microtubules. Thereby, the nucleus is indirectly exposed to external mechanical forces acting in a tensile (via F-actin) and compressive (via microtubules) fashion (**a**), e.g., induced by FSS (**b**). In particular, the filaments are connected to the inner and outer nuclear membrane (INM/ONM) via LINC complexes, composed of the nesprin and SUN proteins (**c**). For microtubule, connection to LINC complexes is mediated via dynein. By transmission of external forces, opening of NPCs might be regulated, allowing a more efficient influx of TFs, like SMADs or F/G-actin-ratio-sensing MRTF, or co-factors like YAP/TAZ. TFs can regulate the transcription of target genes in regions of “open” heterochromatin. SMADs can direct HMEs to target gene sequences, thereby allowing epigenetic modification and thus the activation or repression of genes (**d**). Chromatin compartments of inactive euchromatin are merely found to be proximal to the INM in lamin-associated LADs. Additional to lamins, integral proteins like MAN-1 sit in the NM. MAN-1 confers a further layer of BMP signaling regulation, by binding SMADs and competing with the co-factor binding. Abbreviations: NPC—nuclear pore complex, ONM—outer nuclear membrane, PNS—perinuclear space, INM—inner nuclear membrane, LAD—lamina-associated domain, HME—histone modifying enzyme, and SBE—SMAD binding element.

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
