# Peer review of "It Takes Two to Tango: Endothelial TGFβ/BMP Signaling Crosstalk with Mechanobiology"

_cells, 2020, doi:10.3390/cells9091965_

Round 1
Reviewer 1 Report
The paper proposed by C et al. is a review describing the involvement of the TGF beta /BMP signaling pathway in vascular homeostasis, and the different ways in which the endothelium must integrate mechanical signals to modulate cellular response to these growth factors. This topic is important because the endothelium is a tissue subject to high mechanical stress, and ther is growing evidence that mechanical cues such as shear stress and matrix stiffness are involved in the regulation of BMP signaling.
General comment :
The paper is very dense and long and includes more than 300 references. The abundance of detail and the search for completeness undermines the clarity of the message. The first 4 pages of the article describe in detail the BMP/TGF beta signalling pathways without addressing the specific theme of the review: mechanobiology. This part (2 ; 3) should be shortened/simplified to facilitate reading and focus on mechanobiology. Each paragraph of the review generally provides too much detail and fails to simply conclude on the take home message of the paragraph, which is an important concept in a review.
The figures are numerous and very rich as well, but they are widely under-used in the text: they do not sufficiently represent a support for understanding the text, but additional information to the text (especially figures 1 and 2). The review would strongly benefit the elimination of detail and simpler presentation of concepts, including use/description of figures . Finally, the introduction is rather confusing and lacks structure. It contains also many details but do not clearly set the basis for the content of the review.
Specific comments
- Line 27-28 : The journals that are cited are not adapted or are too old (2009) to deal with the so mentioned mouse models.
- Introduction :Line 38 : The author should first clearly explain what the mean by inside-out or outside in fashion. The term usually refer to integrin signaling but in that context, the notion should be clearly and simply explained.
- Section 2 :this sections deals with an important figure (figure1) with a very complete legend, but the figure is not used/described in the text. It just appears as a illustration.
- Fig 1 : BMP gradient on that context? Reference?
- Line 110-111 : the meaning is not clear (inversion?)- To be corrected
- Line 148 : I don’t understand “This junctional resolution is a hallmark of endothelial-to-mesenchymal transition (EndMT) ». The paragraph deals with quiescence and maturation of vessels and the section on endMT appears irrelevant and should be removed (out of topic).
- Line 153 : In the absence of TGFß/BMP ligands, EC are quiescent : this sentence is wrong : BMP9/10 signaling is responsible for activating quiescence pathways. Quiescence is not the absence of BMP signaling.
- Line 169-170 : The limitation of EC number by BMP9/10 has been demonstrated in the zebrafish. The model cited should be specified since the author then talk about a mouse model. It doesn't seem wise to mix different models that are quite different. Even ref 82 is a zebrafish model and the text refer « in mice ». Since BMP10 isoforms are different between mice and zebrafish, caution should be taken with respect to the conclusions that can be drawn from each of the models.
- Table 1 is of very limited interest.
- The section on the release of latent TGFbeta (4.1) is very long and the general mechanism could be refered to previously published reviews.
Author Response
Dear Reviewers,
With great pleasure, we provide a major revision to the above-mentioned manuscript. First
of all, we thank both reviewers very much for taking their time and for providing valuable
comments which allowed us to improve the manuscript. We have made major edits according
to reviewer 1’s suggestions and we have added details on potential BMP-mechano crosstalk
to TRPV4 as suggested by reviewer 2.
Responses to general comments by reviewer 1:
“The paper is very dense and long and includes more than 300 references. The abundance of
detail and the search for completeness undermines the clarity of the message. The first 4
pages of the article describe in detail the BMP/TGF beta signalling pathways without
addressing the specific theme of the review: mechanobiology. This part (2 ; 3) should be
shortened/simplified to facilitate reading and focus on mechanobiology. “
ð We agree with this concern raised by reviewer 1, particularly general comments
referring to the length and relevance of introduction into TGFβ/BMP signaling in
paragraphs 2 and 3 for the main topic mechanobiology. Since we are combining three
different topics (TGFβ/BMP signaling; Mechanobiology; Endothelial cell functions) this
was meant to allow less experienced audience to understand the combination
thereof. We have extensively shortened and re-written chapters 2, 2.1, 2.2, 2.3 and 3
to facilitate the mechanobiology focus and created instead a concise shortened new
chapter 2 following reviewer 1s request:
2. Activating versus homeostatic TGFβ/BMP signaling in endothelial cells
(New lines 47-147)
Which replaces former chapters:
2. Activating versus homeostatic action of TGFβ/BMP ligands in endothelial cells
2.1. Pro angiogenic BMPs
2.2. BMPs providing maturation and maintaining quiescence
2.3. TGFβ has bipartite role
3. Endothelial SMAD signaling
(Former lines: 89-243)
“Each paragraph of the review generally provides too much detail and fails to simply conclude
on the take home message of the paragraph, which is an important concept in a review.”
ð We respectfully disagree here with reviewer 1’s view. While we do know that many
reviews are presented in a simplistic approach, we have made a conscious decision
here and decided to opt for an alternative way of resuming the existing literature in
detail. In fact, over the course of our literature research, we came across several
original research articles with opposing, sometimes seemingly contradictory results
on the same research question, i.e. effects of FSS on BMP-SMAD signaling, role of
different BMPs in activation of the endothelium. Part of that could be due to
generalization and oversimplification of concepts behind. Instead, we wanted to take
into account that the BMP/TGFβ signalling pathways are intrinsically prone to
feedbacks, dose dependencies, cellular context, crosstalks and switches which the
reviewer perceives as “presented with too much detail”. Thankfully, MDPI review
format is free of space limitations giving us the opportunity to avoid oversimplification
and for carefully respecting the complexity of this intertwined cellular signaling
network. This is why we have chosen for MDPI cells to publish this work. We believe
that biological complexity needs to be shown and presented accordingly to
understand and most importantly better interpret seemingly contradictory data.
ð We still tried to satisfy reviewer 1 by adding more resuming “take home messages” at
the end of each paragraph to further facilitate the reviewing style:
ð Chapter 2: New Lines: 145-147
ð Chapter 3.1: New Lines 375-379
ð Chapter 4: New Lines 470-474
ð Chapter 5.1: New Lines 555-559
“The figures are numerous and very rich as well, but they are widely under-used in the text:
they do not sufficiently represent a support for understanding the text, but additional
information to the text (especially figures 1 and 2).”
ð We very much agree and have inserted more callouts for figure 1 (including subfigures
a,b,c,d,e,f) to better connect it with our newly structured and shortened chapter 2.
“2.2 TGFβ/BMP-SMAD signaling crosstalk to mechanobiology at distinct subcellular
compartments.” (see also answers to specific comments)
ð We have re-edited figure 1 and 2:
Figure 2: (switch of signal-scheme to figure 1)
ð We have further reduced the content of figures:
Figure 2: More focus on compartments
Figure 3: (reduced amount of non-canonical pathway information, reduced pathway
connectivity indicated by arrows, reduced actin cables, reduced ligands and labels)
Figure 4: (simplified Src cycle, reduced tight junction proteins, reduced arrows, reduced
labelling, reduced catenin complex, reduced actin filaments)
Figure 6: (deletion of spectrin ,Golgi, ER, intermediate filaments, labelling)
“The review would strongly benefit the elimination of detail and simpler presentation of
concepts, including use/description of figures .
ð We think that oversimplification is problematic with this topic. We think that we have
provided a clear structure for the review guiding the reader through the different
compartments in which crosstalk appears.
ð We have added further callouts of figures 2-6 and subfigures to better connect all
figures to the text. Figure callouts are listed below:
Figure 2: New line: 220
Figure 2a: New line: 221
Figure 2b: New line: 222
Figure 2b, I: New line: 224
Figure 2b, II: New line: 235
Figure 2b, III: New line: 242
Figure 2b, IV: New line: 252
Figure 3a: New line: 272
Figure 3b: New line: 273
Figure 3c: New line: 277, 319, 324, 332, 337, 355, 356, 360
Figure 4a: New line: 393, 427, 439, 458, 465
Figure 4b: New line: 399, 447
Figure 4c: New line: 399, 447
Figure 5a: New line: 485, 524, 538, 542
Figure 5b: New line: 492
Figure 6a: New line: 574, 608
Figure 6b: New line: 605, 608
Figure 6c: New line: 605, 624, 632, 671, 686
Figure 6d: New line: 640
“Finally, the introduction is rather confusing and lacks structure. It contains also many details
but do not clearly set the basis for the content of the review.”
ð We have done major restructuring and shortening of the introduction to reduce
details. The new introduction sets now the overall structure of the review much better
in our view (former lines: 23-65) New lines: 22-45
Responses to specific comments by reviewer 1
“Line 27-28 : The journals that are cited are not adapted or are too old (2009) to deal with the
so mentioned mouse models.”
ð Endothelial specific/vascular BMP/TGFβ mouse models were more recently
summarized by Goumans and co-workers and we have added this 2018 citation (doi:
10.1101/cshperspect.a031989, [9]) to the references cited in former lines 27-28, New
line 33 hoping this will satisfy reviewer 1. If with “not adapted” is meant “not
complete” we kindly ask for providing the reference that the reviewer thinks is
comprising further relevant endothelial specific BMP/TGFβ mouse models which is not
mentioned in our citations.
“Introduction :Line 38 : The author should first clearly explain what the mean by inside-out or
outside in fashion. The term usually refer to integrin signaling but in that context, the notion
should be clearly and simply explained.”
ð Our view is in line with the field view of integrin-mediated outside-in and inside-out
mechanotransduction. We have re-written this paragraph and shifted it to new
chapter 2.2 when introducing the focal adhesion compartment in order to leave no
room for misinterpretation. We have further added two citations which explain in
depth the above mentioned terms and puts them into context of TGFβ/BMP receptor
signaling (Hao-Luo, Springer et al. 2007 [117]; Munger and Sheppard 2011 [116]).
(New lines 224-230):
“)). Interaction of ECs to ECM (Figure 2b, I) allows for integrin-mediated “outside-in and/or insideout
signaling” [116]. The “outside-in” binding of ECM ligands to cell surface integrins stimulates
conformational changes that induce intracellular signaling by integrin-associated proteins [117].
“Inside-out signaling” strengthens adhesive EC contacts and the appropriate force necessary for
integrin-mediated cell migration, invasion, ECM remodeling, and traction force. Inside-out
integrin-signaling allows cells to e.g. generate traction forces that participate in liberating latent
ECM-bound TGFβ.”
Section 2 :this sections deals with an important figure (figure1) with a
very complete legend, but the figure is not used/described in the text. It
just appears as a illustration.
ð We have re-written the complete legend of figure 1 and shifted information from the
previous figure 1 legend to new chapter 2. We have inserted multiple callouts for
figure 1 and each sub-figure (a-f) in chapter 2 to better integrate this illustration. We
have complemented figure 1 with previous figure 2c (now figure 1f) due to the
restructuring.
ð Figure 1 citations:
Figure 1a : New lines 51, 53, 57, 79, 93
Figure 1b: New line 54
Figure 1c : New lines 82, 139
Figure 1d: New lines 84, 89, 139
Figure 1e: New line 103, 139
Figure 1f: New lines: 121, 174, 189
Fig 1 : BMP gradient on that context? Reference?
ð In-vivo, such gradient can only be proposed, based on extensive in-vitro studies by
multiple labs including ours using chemotactic chamber assays. We have made this
hypothesis now more clear by:
1. Inserting question marks into the graphical gradient (Figure 1a, upper, blue)
2. Adding a new explanation to pro-angiogenic gradients clearly describing that this
gradient is so far speculative in-vivo:
New lines 69-73:
“Whether EC activating BMP2,6,7 gradients exists in-vivo is still debatable. While BMP
gradients are well described in early developmental tissue patterning of invertebrates and
vertebrates [46], their existence and contribution for sprouting angiogenesis in-vivo is still not
clearly shown. Mouse data on BMP-induced tumor vascularization however suggest, that
BMPs induce tumor angiogenesis similarly to vascular endothelial growth factor (VEGF)-like
gradients [39,40].”
3. Depicting this clearly in the figure caption (New lines: 155-156):
“Angiogenic gradients of activating BMPs 2/6/7 (blue) are proposed based on in-vitro
data but not shown to date to exist in-vivo.”
“Line 110-111 : the meaning is not clear (inversion?)- To be corrected”
ð We have re-written this sentence in the light of new chapter 2: to “Tip-cell filopodia
promote sprouting …” (New lines 61-62)
Line 148 : I don’t understand “This junctional resolution is a hallmark of endothelial-tomesenchymal
transition (EndMT) ». The paragraph deals with quiescence and maturation of
vessels and the section on endMT appears irrelevant and should be removed (out of topic).
ð We agree that this paragraph was mislocated and have moved it to chapter 4
(BMP/TGFβ signaling and integration of basolateral forces), (now lines 383-388). We
would like maintaining it within the review since in our view, alterations in TGFβ/BMP
signaling leading to EndMT are tightly coupled to EC junctional resolution.
Line 153 : In the absence of TGFß/BMP ligands, EC are quiescent : this sentence is wrong :
BMP9/10 signaling is responsible for activating quiescence pathways. Quiescence is not the
absence of BMP signaling.
ð This sentence (former line 153) reads: “In the absence of activating TGFβ/BMP
ligands, ECs are quiescent….”. Meaning absence of intraluminally delivered BMP9/10
ligands but only presence of extracellularly delivered BMP2,6,7 ligands that have
activating e.g. pro-angiogenic functions on the endothelium. Such is possible in a
newly formed angiogenic sprout that are not transfused with blood yet. However, due
to restructuring of this chapter, this sentence is now deleted from the manuscript.
“Line 169-170 : The limitation of EC number by BMP9/10 has been demonstrated in the
zebrafish. The model cited should be specified since the author then talk about a mouse
model. It doesn't seem wise to mix different models that are quite different. Even ref 82 is a
zebrafish model and the text refer « in mice ». Since BMP10 isoforms are different between
mice and zebrafish, caution should be taken with respect to the conclusions that can be drawn
from each of the models.”
ð We agree that mixing conclusions from different model systems without specifying
them is problematic and we apologize for mentioning the wrong system referring to
reference 82 (now 66). We have specified the model systems cited accordingly and
added a sentence on that context:
New lines 103-114:
“In zebrafish, it was shown that BMP9/10-Alk1 signaling limits EC numbers and, thereby, stabilizes
the caliber of nascent arteries [65]. Additionally, Alk1 expression depends on fluid shear stress (FSS)
exerted by blood flow in zebrafish [66] and some flow-responsive genes are dysregulated in Alk1
mutant arterial ECs, suggesting Alk1 to be the main BMP type I receptor integrating endothelial FSS
into biochemical signaling responses [66]. Furthermore, deletion of ALK1 in mice leads to exuberated
sprouting in the mouse retina [16] and addition of BMP9 normalized aberrant tumor vasculature by
decreasing permeability in Lewis lung carcinoma (mice) [67]. Studies using human cells revealed that
BMP9-induces expression and secretion of stromal cell-derived factor 1 (SDF1/ CXCL12) which
promotes vessel maturation by regulating mural cell coverage [68] and counteracts VEGF-induced
angiogenesis [59]. However, comparison of different model systems for Bmp10-Alk1 signaling should
be taken with care due to the very different nature of vascular beds, flow regimes and paralog
expression [69].”
“Table 1 is of very limited interest.”
ð We have removed table 1 from the manuscript, which, together with the
restructuring, has also led to removal of 18 citations from the reference list.
ð We would like to mention that we have added one more relevant citation which has
been published recently on a pre-print server:
“Circulating BMP9 protects the pulmonary endothelium during inflammation-induced
lung injury in mice; Wei Li …. Nicholas W Morrell et al. 2020” (ref. [194])
“The section on the release of latent TGFbeta (4.1) is very long and the general mechanism
could be refered to previously published reviews.”
ð To our knowledge, this is the first review suggesting endothelial traction forces beeing
involved in the liberation of latent TGFβ, based on previous data by us. The fact that
latent TGFβ is mechanically released is not very well distributed within the vascular
reseach community, as we have learned from recent lectures listened to or given in
front of international scientists.
ð Thus, former chapter 4.1 (now 3.1) provides a major conceptual novelty that should
be explained in detail for readers of the endothelial cell community who are reading
this review. We agree that there are numerous excellent reviews by others on latent
TGFβ liberation by integrin-mediated traction forces but these reviews focus on cell
types other than endothelial cells, e.g. fibroblasts and typically refer to tissue fibrosis
not endothelial function. Thus, we would like to maintain this paragraph in its current
form.
Reviewer 2 Report
This is a comprehensive and clear review of the field of BMP/TGFB interaction with mechanobiology in endothelial cells. It covers all relevant areas and provides in-depth information. It will be a valuable resource for investigators in the field.
The only additional area that I could suggest discussing is the interaction of calcium signaling with TGFB/BMP pathways. As briefly mentioned in the manuscript, mechanosensitive calcium channels are an important player in sensing of shear stress in endothelial cells. TRPV4 are likely the most important (Hartmannsgruber V, Heyken WT, Kacik M, Kaistha A, Grgic I, Harteneck C, Liedtke W, Hoyer J, Kohler R. Arterial response to shear stress critically depends on endothelial TRPV4 expression. PloS One 2: e827, 2007.). Of relevance to this manuscript, TRPV4 are also important in angiogenesis:
Thoppil, R. J., Cappelli, H. C., Adapala, R. K., Kanugula, A. K., Paruchuri, S., and Thodeti, C. K. (2016). TRPV4 channels regulate tumor angiogenesis via modulation of Rho/Rho kinase pathway.
Adapala, R. K., Thoppil, R. J., Ghosh, K., Cappelli, H. C., Dudley, A. C., Paruchuri, S., et al. (2016). Activation of mechanosensitive ion channel TRPV4 normalizes tumor vasculature and improves cancer therapy.
Although speculative, it is possible that TRP channels and TGFbeta interact in endothelial cells in a similar fashion to fibroblasts:
Adapala RK, Thoppil RJ, Luther DJ, et al. TRPV4 channels mediate cardiac fibroblast differentiation by integrating mechanical and soluble signals. J Mol Cell Cardiol. 2013;54:45-52. doi:10.1016/j.yjmcc.2012.10.016
Author Response
Please find point-by-point response to comments by reviewer 2 below:
This is a comprehensive and clear review of the field of BMP/TGFB interaction with
mechanobiology in endothelial cells. It covers all relevant areas and provides in-depth
information. It will be a valuable resource for investigators in the field.
ð Thank you very much for this encouraging evaluation of our work.
The only additional area that I could suggest discussing is the interaction of calcium signaling
with TGFB/BMP pathways. As briefly mentioned in the manuscript, mechanosensitive calcium
channels are an important player in sensing of shear stress in endothelial cells. TRPV4 are
likely the most important (Hartmannsgruber V, Heyken WT, Kacik M, Kaistha A, Grgic I,
Harteneck C, Liedtke W, Hoyer J, Kohler R. Arterial response to shear stress critically depends
on endothelial TRPV4 expression. PloS One 2: e827, 2007.). Of relevance to this manuscript,
TRPV4 are also important in angiogenesis:
Thoppil, R. J., Cappelli, H. C., Adapala, R. K., Kanugula, A. K., Paruchuri, S., and Thodeti, C. K.
(2016). TRPV4 channels regulate tumor angiogenesis via modulation of Rho/Rho kinase
pathway.
Adapala, R. K., Thoppil, R. J., Ghosh, K., Cappelli, H. C., Dudley, A. C., Paruchuri, S., et al. (2016).
Activation of mechanosensitive ion channel TRPV4 normalizes tumor vasculature and
improves cancer therapy.
Although speculative, it is possible that TRP channels and TGFbeta interact in endothelial cells
in a similar fashion to fibroblasts:
Adapala RK, Thoppil RJ, Luther DJ, et al. TRPV4 channels mediate cardiac fibroblast
differentiation by integrating mechanical and soluble signals. J Mol Cell Cardiol. 2013;54:45-
52. doi:10.1016/j.yjmcc.2012.10.016
ð We agree that the interplay of TGFβ/BMP receptors with ion-channels, particularly
TRPV4, due to the clear connection between TGFβ/BMP signaling and endothelial
calcification (e.g. in arteriosclerosis), as well as FSS related changes in cell volume is
likely to occur. A potential crosstalk of TGFβ/BMP signaling with TRPV4 at the ciliary
compartment should indeed be studied in more detail.
ð We have added the following paragraph in chapter 6 line 521 (New lines 494 -501):
“Interestingly, depletion of osmolarity and cell volume regulating calcium channel TRPV4 in mice
completely abolishes shear stress induced vasodilation [213]. TRPV4 localizes to mesenchymal
stem cell cilia [214] and it was shown that the differentiation process of cardiac fibroblasts to
myofibroblasts is dependent on TRPV4 which integrates mechanical (i.e. ECM stiffness) and TGFβ
signals [215]. Moreover, interfering with TRPV4 signaling blocks TGFβ induced epithelial-tomesenchymal
transition (EMT) in normal mouse primary epidermal keratinocytes (NMEKs) [216].
It is thus tempting to speculate that TRPV4 may also be a regulator of mechano-TGFβ crosstalk in
the endothelium.”
Round 2
Reviewer 1 Report
The manuscript has been significantly improved. Modifications bring a much clearer point of view of BMP9/TGFß signaling regulation. Thanks to the authors for this substantial revision work.
In line to the recent paper added : Li et al., circulating BMP9 protects the pulmonary endithelium during inflammation-induced lung injury in mice, the recent paper of Szulcek et al. in angiogenesis "exacerbated inflammatory signaling underlies aberrant response to BMP9 in pulmonary arterial hypertension lung endothelial cells" could be mentioned.